# Patient Dose Estimation in Computed Tomography-Guided Biopsy Procedures

**DOI:** 10.3390/jimaging9120267

**Published:** 2023-11-30

**Authors:** Evangelia Siomou, Dimitrios K. Filippiadis, Efstathios P. Efstathopoulos, Ioannis Antonakos, George S. Panayiotakis

**Affiliations:** 1School of Health Sciences, University of Patras, 26504 Patras, Greece; esiomou@med.uoa.gr (E.S.); panayiot@upatras.gr (G.S.P.); 22nd Department of Radiology, National and Kapodistrian University of Athens, 1st Rimini St., Chaidari, 12461 Athens, Greece; dfilippiadis@med.uoa.gr (D.K.F.); stathise@med.uoa.gr (E.P.E.)

**Keywords:** CT-guided biopsies, diagnostic reference levels, patient dose

## Abstract

This study establishes typical Diagnostic Reference Levels (DRL) values and assesses patient doses in computed tomography (CT)-guided biopsy procedures. The Effective Dose (ED), Entrance Skin Dose (ESD), and Size-Specific Dose Estimate (SSDE) were calculated using the relevant literature-derived conversion factors. A retrospective analysis of 226 CT-guided biopsies across five categories (Iliac bone, liver, lung, mediastinum, and para-aortic lymph nodes) was conducted. Typical DRL values were computed as median distributions, following guidelines from the International Commission on Radiological Protection (ICRP) Publication 135. DRLs for helical mode CT acquisitions were set at 9.7 mGy for Iliac bone, 8.9 mGy for liver, 8.8 mGy for lung, 7.9 mGy for mediastinal mass, and 9 mGy for para-aortic lymph nodes biopsies. In contrast, DRLs for biopsy acquisitions were 7.3 mGy, 7.7 mGy, 5.6 mGy, 5.6 mGy, and 7.4 mGy, respectively. Median SSDE values varied from 7.6 mGy to 10 mGy for biopsy acquisitions and from 11.3 mGy to 12.6 mGy for helical scans. Median ED values ranged from 1.6 mSv to 5.7 mSv for biopsy scans and from 3.9 mSv to 9.3 mSv for helical scans. The study highlights the significance of using DRLs for optimizing CT-guided biopsy procedures, revealing notable variations in radiation exposure between helical scans covering entire anatomical regions and localized biopsy acquisitions.

## 1. Introduction

Computed Tomography (CT) has become a cornerstone in the realm of diagnostic imaging, playing a pivotal role in the diagnosis and management of a wide range of medical conditions. Its widespread use, however, has been accompanied by concerns over the relatively high radiation doses delivered to patients, which, when accumulated over time, have the potential to substantially elevate the risk of radiation-induced cancers. In response to these concerns, the medical community has sought innovative and less invasive ways to accurately assess potential malignancies, particularly in the lung, bone, and abdominal regions [1]. This has led to the emergence of CT-guided biopsies, a technique that not only minimizes the risk of complications but also offers the benefit of quicker recovery times and obviates the need for general anesthesia, providing a more patient-friendly approach to diagnosis [2].

CT-guided biopsies are typically executed using the coaxial needle technique, a well-established method that provides a stable and precise position for the biopsy procedure. The coaxial needle technique has garnered widespread acceptance and routine use in radiological practice, largely due to its reliability and efficacy. Operators are increasingly confident in its application, as advances in technology continue to enhance its safety and performance. The result is a diagnostic approach characterized by a high degree of sensitivity, specificity, and diagnostic accuracy, enabling the distinction between benign and malignant lesions [3].

Moreover, ongoing progress in CT imaging technology has offered a dependable means of visualizing lesions and the intricate surrounding anatomical structures. These developments empower medical practitioners with a clearer and more comprehensive understanding of a patient’s condition, enabling more informed decision making. The amalgamation of effective image guidance and state-of-the-art biopsy devices further refines this procedure, turning it into a critical tool in modern healthcare practice [4].

Interventional CT procedures encompass a diverse array of scan modes, with the choice of mode contingent upon factors such as the specific imaging objectives, scanner setup, and operator preferences. Among these modes, CT fluoroscopy stands out for its real-time capabilities, involving the continuous acquisition of data that are promptly reconstructed to offer instantaneous feedback to the operator [5,6]. This dynamic approach is complemented by versatile patient table options, which may remain stationary or be manually repositioned by the operator, thereby affording flexibility in patient positioning and alignment. An alternative technique involves the acquisition of discrete axial images, typically generated one per step on the pedal and coordinated with the movements of both the patient table and the interventional instruments [5,7]. However, certain interventional procedures, such as cryoablation, necessitate the use of helical scans due to the imperative need to monitor the anatomical region along the z-axis, often extending beyond the detector width of conventional CT systems. Helical scans, therefore, become indispensable, with imaging sequences repeated at various time points, including the initial localization scan, during the procedure itself, and immediately post-procedure, ensuring continuous and comprehensive visualization of the targeted region.

Recent advancements in radiation safety guidelines have been introduced to address the critical issues of dose justification, optimization, and limitation, with a particular emphasis on medical exposure, especially in the context of CT imaging. These guidelines are provided by two significant regulatory bodies: the International Basic Safety Standards (BSS) and the latest European Directive 2013/59/Euratom on radiation protection [8,9]. A noteworthy aspect of these standards is their requirement for the maintenance of radiation dose data within patient examination records, particularly concerning CT and interventional systems. This stipulation ensures that radiologists have immediate access to essential dose information during interventional procedures, enabling them to make real-time decisions that prioritize patient safety. Additionally, these directives represent a pivotal development in radiation safety by introducing a new set of standards that address the registration and analysis of unintentional and accidental medical exposures, thus fostering a comprehensive and proactive approach to managing radiation exposure incidents in the medical field [10].

The concept of Diagnostic Reference Levels (DRLs) holds paramount significance in both European [11] and international [12] radiological communities as a significant tool for optimizing the administration of medical imaging procedures with a keen focus on minimizing patient radiation exposure. This approach is not only consistent with the recommendations of the International Commission on Radiological Protection (ICRP) in terms of optimizing X-ray examinations [12] but has also evolved to become an imperative facet of contemporary radiology practices [11]. The ICRP, in its most recent guidance in 2017, further underscores the importance of DRLs, with a dedicated section on Diagnostic Reference Levels in medical imaging that specifically addresses the context of CT-guided interventional procedures.

The primary objective of this study was to comprehensively examine and quantify the radiation burden borne by patients undergoing CT interventional procedures, with a specific focus on two critical dosimetric parameters: the Entrance Skin Dose (ESD) and the Effective Dose (ED). This investigation sought to provide a thorough assessment of the potential risks associated with these procedures and contribute valuable insights into radiation safety measures. Furthermore, the study aimed to establish typical Diagnostic Reference Levels (DRLs) using the Computed Tomography Dose Index (CTDI_vol_) and the Dose-Length Product (DLP) as reference points. These typical DRL values, once derived, can serve as a fundamental tool for defining regional or national DRLs where such standards may be currently lacking, thereby promoting a more standardized and uniform approach to radiation dose management in CT interventional procedures. The creation of these DRLs is instrumental not only in enhancing patient safety but also in ensuring that medical practitioners consistently adhere to well-defined benchmarks in radiation exposure, fostering a culture of continuous optimization and adherence to international standards.

## 2. Materials and Methods

This study retrospectively analysed a total of 226 CT-guided needle biopsies, spanning a five-year period from January 2017 to October 2022. These procedures were carried out at the Radiology Department of “ATTIKON” University Hospital in Greece, an institution renowned for its commitment to medical research and the delivery of high-quality patient care. The patient cohort under investigation was notably diverse, comprising 163 male and 63 female patients, collectively reflecting a wide spectrum of clinical presentations, gender diversity, and age demographics. Notably, the patients encompassed a broad age range, spanning from 17 to 88 years, with a mean age of 65 years, underlining the heterogeneity and inclusivity of individuals who underwent these CT-guided needle biopsies. This age diversity underscores the wide-reaching relevance of such interventional procedures across different age groups and highlights their significance in the context of cancer diagnosis and disease management.

Moreover, the patients included in the study exhibited varying weights within the range of 70 to 80 kg, signifying a diversity in body mass index and body habitus, factors that can significantly influence the technical aspects of CT-guided biopsies, including equipment settings, patient positioning, and overall procedural planning. This, in turn, emphasizes the relevance of this research in optimizing patient-specific care and tailoring interventional approaches to individual patient characteristics. The biopsy samples were thoughtfully categorized into distinct groups, with a total of 90 cases located within the thoracic cavity, comprising 57 cases within the lung and 33 within the mediastinum.

Additionally, 57 biopsies were conducted within the liver, 48 within the musculoskeletal system, particularly the iliac bone, and 31 within the para-aortic lymph nodes. This comprehensive categorization underscores the broad range of anatomical locations and organ systems that were subject to biopsy and exemplifies the thorough scope of this investigation. The clinical indication for each of these biopsies was the presence of a focal lesion, nodule, or mass suspected to be malignant, indicating the vital role played by CT-guided needle biopsies in the detection, evaluation, and diagnosis of potentially malignant lesions across diverse anatomical sites.

During the CT-guided needle biopsy procedures, patients were positioned on the CT gantry, and the precise skin location above the intended puncture site was clearly demarcated using a radiopaque substance. The selection and execution of the needle path, adhering to strict sterile techniques, constituted a critical phase of the biopsy procedure. To ensure the accurate introduction, progression, and placement of the biopsy needle, intermittent CT scans were diligently obtained throughout the procedure, employing the specialized biopsy mode. This mode entailed image acquisition via the short helical scanning method, offering real-time visualization and guidance. Importantly, these CT-guided biopsy procedures were exclusively conducted by a highly skilled and certified interventional radiologist, who was expressly instructed to limit the preprocedural planning scan length to enhance the efficiency and precision of the process. The biopsies were carried out utilizing a state-of-the-art 64-MDCT scanner, specifically the Brilliance 64 model by Phillips, which ensured high-quality imaging and accurate guidance throughout the procedures. The acquisition protocols for each distinct biopsy procedure were thoughtfully designed and customized, a testament to the attention to detail and the rigorous planning that underpinned this comprehensive investigation, as summarized in Table 1.

Using the CT DICOM data, dosimetric indices (DLP and CTDI_vol_, referred to as the standardized 32 cm polymethyl-methacrylate body phantom) were retrospectively collected for analysis.

The Size-Specific Dose Estimate (SSDE) and the Effective Dose (ED) were derived using the appropriate k-factors and conversion factors proposed by the ICRP and AAPM, respectively, CTDI_vol_ reflects the average radiation exposure per section, and DLP reflects the total radiation output for the examination [11,13]. SSDE indicates the doses at the center of the scanned region of an individual patient, considering the patient’s size. The ED is the tissue-weighted sum of the equivalent doses in all tissues and organs and can be used to estimate the stochastic health risk of cancer induction. These values were separated based on the entire anatomic area scan protocol and repeated biopsy scans to investigate the radiation burden.

The estimation of skin dose was conducted for each individual procedure, adhering to a defined formula [5]. It is worth emphasizing that throughout the entirety of this study, all CT scans were conducted in strict accordance with the helical scan mode.
(1)$$skindose=\left\{\begin{array}{c} 1.2\times CTDIvol,\;\;for\;helical\;mode\\ 0.6\times CTDIvol,\;\;for\;intermittent\;mode \end{array}\right.$$

The ICRP recommends that the DRLs for CT-guided procedures be set in terms of the CTDI_vol_, number of sequences, and CT fluoroscopy time in the case of CT fluoroscopy-guided techniques. The report also defined the term “typical DRLs” as the median of the distribution of data for a DRL quantity for a clinical imaging procedure. The median, min, max, and 1st and 3rd quartiles values of the distributions of all DLP and CTDI_vol_ values were also calculated for benchmarking purposes using the literature.

## 3. Results

The study included a total of 226 biopsy procedures performed across five distinct CT interventional protocols, targeting anatomical sites including the iliac bone, liver, lung, mediastinal mass, and para-aortic lymph nodes. To comprehensively evaluate the dosimetric aspects and the associated radiation exposure, a range of essential dosimetric quantities, including DLP, CTDI_vol_, SSDE, and ED, were systematically examined. To provide a detailed and comprehensive overview of the findings, these critical dosimetric parameters were thoughtfully tabulated in Table 2, Table 3, Table 4, Table 5 and Table 6, each containing key statistics such as mean values, standard deviations, medians, minimum and maximum values, and the 1st and 3rd quartiles for each of the specific CT-guided biopsy procedures, distinguishing between helical and biopsy scan modes. This data-rich presentation enables healthcare professionals and researchers to access and interpret the variations in radiation exposure, ultimately contributing to a deeper understanding of radiation safety and optimization in the context of CT-guided biopsy procedures across various anatomical regions.

Figure 1, Figure 2, Figure 3 and Figure 4 in the study illustrate the essential statistical insights encapsulated in the five-number summary, which comprises the minimum, first quartile, median, third quartile, and maximum values. These summaries pertain to the distributions of total DLP and CTDI_vol_ for each distinct category of procedures, specifically examining helical and biopsy scans. Notably, the examination of para-aortic lymph nodes biopsies stands out as having the highest recorded values for both total DLP and CTDI_vol_. Total DLP values span a range from 672.4 mGy cm observed in CT-guided iliac bone procedures to 1182.6 mGy cm in CT-guided para-aortic lymph nodes biopsies. These statistical findings serve as valuable insights into the radiation doses associated with different procedural categories, shedding light on the variability and extremities within the dataset, thereby contributing to a more comprehensive understanding of radiation exposure in clinical practice.

In Figure 5, Figure 6, Figure 7, Figure 8 and Figure 9 of the study, an informative depiction is provided regarding the burden percentage associated with both helical and biopsy scans across various categories of biopsy procedures, focusing on both ESD and ED. Specifically, these percentages offer insights into the distribution of the radiation burden within the dataset. In the helical mode, the ESD burden ranges from 53% to 64% across the different categories of biopsy procedures. In parallel, when considering biopsy scans for all cases of CT-guided biopsies, the ESD percentages exhibit a range from 36% to 47%. In terms of the ED, the percentages span from 54% to 80% within the helical mode and from 20% to 46%, respectively, for biopsy scans. These findings illuminate the varying degrees of radiation exposure within the different biopsy procedure categories, underscoring the significance of optimizing radiation doses to minimize patient risk while maintaining diagnostic efficacy in clinical practice.

Figure 10 provides the Typical DRL values denoted in terms of CTDI_vol_ (mGy) for two distinct scan modes, namely helical scans and biopsy scans employed in CT-guided biopsy procedures. Typical DRLs were established based on the rounded values extracted from the median CTDI_vol_ following the guidelines delineated in ICRP 135. For helical mode scans, these indicative DRLs are set within a range spanning from a minimum of 7.9 mGy to a maximum of 9.7 mGy, while for biopsy scans conducted in the context of CT-guided biopsies, the recommended DRLs exhibit a range extending from a lower limit of 5.6 mGy to an upper limit of 7.7 mGy. These meticulously determined reference levels serve as essential tools for the purpose of diligent monitoring and optimization of radiation doses during such radiological examinations. This, in turn, contributes significantly to the overarching goals of patient safety and the minimization of unnecessary exposure to ionizing radiation in clinical practice.

## 4. Discussion

Within the context of this study, DRLs have been recognized as invaluable instruments for the enhancement and fine-tuning of clinical practice, focusing on two pivotal aspects: optimizing patient radiation doses and elevating the overall quality of image acquisition. Notably, these typical DRLs were set by relying on the median Computed CTDI_vol_ values, aligning with the guidance set forth in the ICRP 135 report [14]. It is worth noting that the research also embraced a broader perspective by considering additional statistical parameters, such as the mean, median, as well as the 1st and 3rd quartiles of DLP values. These additional metrics were calculated with the intent of facilitating comparisons with existing literature, thereby contributing to a more comprehensive and robust evaluation of radiological practices. This multi-pronged approach underscores the study’s commitment to not only establishing benchmark DRLs but also to contextualizing them within a broader framework of dose and image quality assessment.

The results obtained from this study were subject to a rigorous comparative analysis, juxtaposed against findings reported in other contemporary research endeavors published since 2011, as summarized in Table 7. It is noteworthy that, to the best knowledge of the authors, the sole multicenter investigation conducted in this domain is the comprehensive study conducted by Greffier et al., who put forth Diagnostic Reference Levels (DRLs) tailored to the French context for interventional CT procedures. In their study, National DRLs were thoughtfully proposed across 17 distinct categories of procedures, encompassing six for thoracic and abdominopelvic procedures, as well as eleven for osteoarticular procedures. The values they presented were derived from rounded figures representing the 3rd quartile of the overall dose distribution within each specific procedure [15,16]. It is important to underscore that other notable studies, such as the one conducted by Leng et al., established DRLs by considering the mean DLP values for five different procedures across a substantial cohort of 571 patients, employing eight CT scans [5]. Similarly, Kloeckner et al. contributed to the DRL framework by determining DLP-based reference levels for 12 procedures, drawing from a dataset of 1284 patients and one CT scan [17]. However, it is essential to note that these earlier investigations were single-center studies and thus may not comprehensively represent the diverse spectrum of national radiological practices, making the current multicenter study a significant and distinctive contribution to this critical field of research.

The DLP values obtained in this study were observed to surpass the 3rd quartile values proposed by Weir et al. specifically for the purpose of tumor ablation in thoracic or abdominopelvic regions [18]. It is important to note, however, that Weir’s study was conducted within the confines of a single medical center for each CT scan, and the DRLs they put forward primarily reflected localized practices, lacking the consideration of the substantial variabilities that have been previously highlighted. Furthermore, when compared to radiation dose levels established in other studies following an optimization process, the DRLs established in this current investigation were notably higher than the mean, median, or 3rd quartile values proposed in these alternative research studies [15,16,19,20,21,22,23]. This differential could be attributed to the multifaceted nature of the clinical variables and practices across different regions and medical centers, underscoring the need for nuanced and context-specific DRLs to ensure prudent and tailored radiation safety measures in diverse clinical settings [24].

The findings in this study have effectively underscored the significant variability in radiation doses administered to patients, a variability that is intricately linked to the specific interventional CT procedures conducted within the same medical center. These variations can be attributed to a multitude of factors, but the paramount influence is exerted by the patients themselves, as the doses delivered are profoundly influenced by their unique anatomical attributes, morphological characteristics, and underlying pathologies.

Notably, this research has highlighted a noteworthy correlation between the number of helical acquisitions and the radiation burden imposed on patients. The utilization of the helical mode, which can be employed both before the procedure (for localization), during it, and at its conclusion (for control), necessitates careful optimization efforts aimed at minimizing its use whenever possible and substituting it with sequential or fluoroscopic acquisition techniques, as previously recommended [16,17,18,19,20]. Furthermore, it is essential to acknowledge that helical acquisition inherently involves a heightened exposure to Z-overscan. In this study, it was observed that the number of helical acquisitions escalated in tandem with the complexity of the procedure, with two acquisitions for iliac bone, mediastinum, and para-aortic lymph node biopsies, and three for liver and lung biopsy procedures. Significantly, the helical mode contributed the most to the total DLP, accounting for an average of 60% for iliac bone, 58% for liver, 80% for lung, 60% for mediastinal mass, and 54% for para-aortic lymph node biopsies. This aligns with the findings reported by Leng et al., who similarly noted substantial proportions ranging from 65% for cimentoplasty to 98% for cryoablation [5]. These results underscore the imperative of judiciously managing helical acquisitions in interventional CT procedures, considering both patient safety and optimization of radiation exposure.

Paik et al. conducted investigations that yielded significant achievements in terms of substantial dose reductions by opting to replace helical acquisitions with fluoroscopy during cervical and lumbar transforaminal epidural injection procedures [21,22]. This strategic shift towards fluoroscopy, as opposed to helical acquisitions, proved to be an effective means of minimizing patient radiation exposure. Furthermore, Greffier et al. observed similar encouraging outcomes in their research involving various osteoarticular procedures [16]. Notably, in addition to advocating for the replacement of fluoroscopy mode with sequential acquisition, their findings lent support to the efficacy of such substitution in achieving reductions in radiation doses. These studies collectively underline the tangible benefits of adopting alternative acquisition modes, such as fluoroscopy and sequential acquisition, as prudent strategies for mitigating patient radiation exposure during specific interventional procedures.

This study represents a pivotal step in the broader endeavor to establish and propose DRLs for a range of interventional CT procedures by leveraging dosimetric data. However, it is essential to recognize that there remain several avenues for further research to comprehensively explore and understand the intricate dynamics of radiation dose optimization in clinical practice. Future studies should delve into the influence of multiple factors on the total DLP, including the number of treatment targets, the specific organ being treated (e.g., hepatic tumor versus renal tumor), diverse anatomical positions within the same procedure (e.g., cervical, thoracic, or lumbar for vertebroplasty), variations in tumor destruction techniques (such as radiofrequency, microwave, or cryotherapy), and the integration of complementary imaging systems alongside CT [25,26,27,28,29,30].

An interesting task is also the correlation of patient dose from CT-guided percutaneous transthoracic needle biopsies of lung lesions with lesions size and depth, as well as procedure time and patient pain reported [31,32,33].

These investigations would provide a more nuanced and holistic understanding of the variables affecting radiation dose in interventional CT procedures, allowing for the development of even more refined and context-specific DRLs.

It is important to acknowledge the limitations of this current study, including the relatively small patient sample size (although it is adequate as proposed by ICRP 135) and the reliance on a single imaging modality, which necessitates continued research efforts to further enhance our knowledge of radiation dose management in clinical practice [31]. Relevant literature supports the importance of addressing these limitations, such as studies by Kim et al. [34], Sangha et al. [35], Cardella et al. [36], and Fu et al. [37] that provide insights into various aspects of CT-guided lung biopsy, including radiation doses, diagnostic performance, and protocol optimization.

**Table 7 jimaging-09-00267-t007:** Comparison of median DLP_total_ values with the recent international literature.

	Present Study (DLP mGy cm)		Other Studies (DLP mGy cm)					
	mean	median	mean	median	mean	median	mean	median
Iliac bone	672.4	489.4			793.0 [24]	113.8 [16]		410.0 [25]
Liver	967.5	748.7	712.0 [24]	652.0 [24]	813.0 [15] 1539.2 [18]	652.0 [17]		710.0 [25]
Lung	894.0	817.2	507.0 [24]	481.0 [24]	440.0 [15] 4320.5 [18]	481.0 [17] 113.8 [20]		435.0 [25]
Mediastinum	744.6	547.2	549.0 [24]	468.0 [24]	440.0 [15] 4320.5 [18]	481.0 [17] 113.8 [20]		435.0 [25]
Para-aortic tissue	1182.6	902.8	781.0 [24]	723.0 [24]	813.0 [15] 1539.2 [18]	652.0 [20]		710.0 [25]

## 5. Conclusions

In this study, the authors conducted a thorough assessment of radiation doses administered during a diverse range of CT-guided procedures. This research endeavor was fundamentally underpinned by the imperative of aligning with the latest European and International regulations governing the safe and responsible utilization of ionizing radiation in the realm of medical imaging. The impetus for this initiative was driven by the escalating demand for robust radiation protection practices, particularly in the context of high-dose procedures like CT-guided techniques, which demand heightened scrutiny. Furthermore, this scientific inquiry was motivated by the conspicuous absence of well-defined regional or national Diagnostic Reference Levels (DRLs) that are specifically tailored to these procedures.

As such, the study was not only geared towards expanding the understanding of radiation exposure in CT-guided biopsy procedures but also sought to bridge this critical gap by meticulously defining typical DRLs. These reference levels, informed by the most up-to-date recommendations and guidelines from the International Commission on Radiological Protection (ICRP), were devised with precision. They were applied to a selection of CT-guided biopsy procedures that spanned a spectrum of anatomical regions, including the iliac bone, liver, lung, mediastinal mass, and para-aortic lymph nodes. In essence, this study represents a significant stride in advancing the field of radiation protection, contributing to the optimization of CT-guided procedures and the enhancement of patient safety, particularly in the context of cancer diagnosis and management.

It is noteworthy that the magnitude of radiation dose imparted to the patient was found to be notably influenced by the repetition of helical scans. However, the investigation also recognized the multifaceted nature of this relationship, acknowledging that a myriad of clinical factors, including the clinical task at hand, the specific tumor site, procedural complexity, patient anatomy, as well as the skill and experience of the attending physician, collectively contribute to the overall radiation dose in CT-guided procedures. This holistic approach underscores the study’s commitment to comprehensively address the intricate interplay between radiation exposure and various clinical variables, thereby enhancing the foundation for evidence-based radiation safety practices in these medical interventions.

## Figures and Tables

**Figure 1 jimaging-09-00267-f001:**
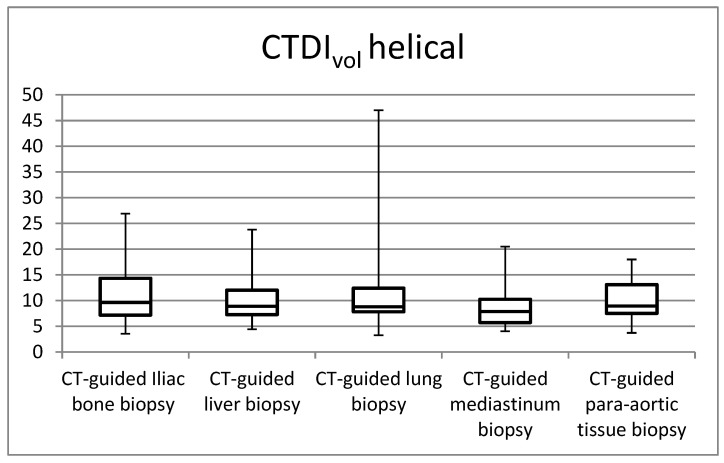
Box plots of the distribution of CTDI_vol_ values for CT-guided procedures at helical acquisitions.

**Figure 2 jimaging-09-00267-f002:**
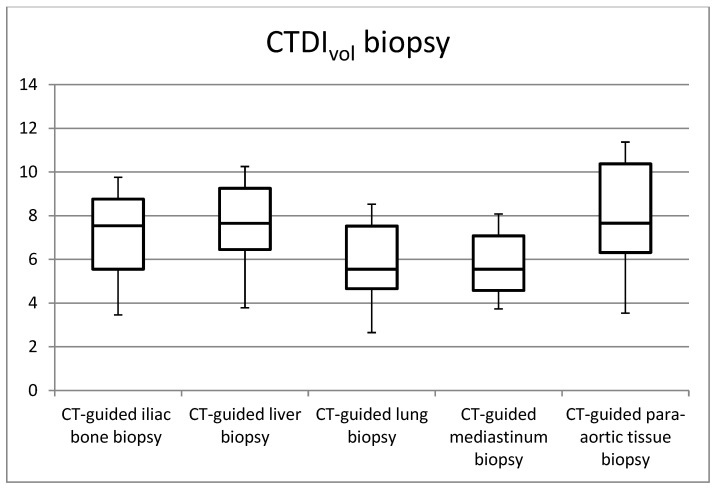
Box plots of the distribution of CTDI_vol_ values for CT-guided procedures at biopsy acquisitions.

**Figure 3 jimaging-09-00267-f003:**
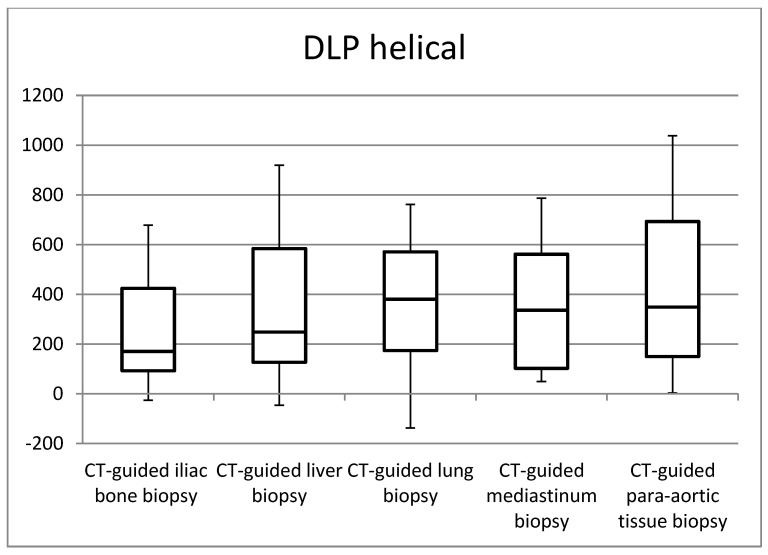
Box plots of the distribution of DLP values for CT-guided procedures at helical acquisitions.

**Figure 4 jimaging-09-00267-f004:**
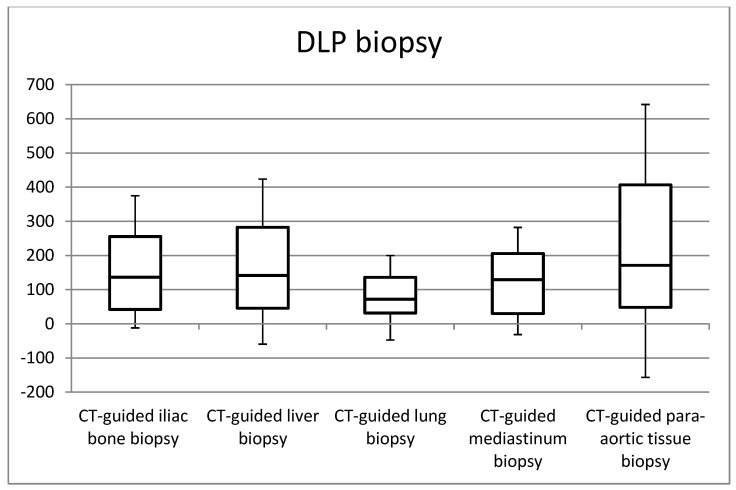
Box plots of the distribution of DLP values for CT-guided procedures at biopsy acquisitions.

**Figure 5 jimaging-09-00267-f005:**
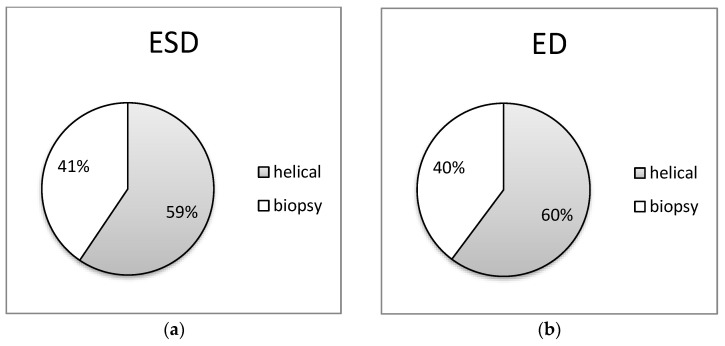
Distribution of (**a**) ESD (mGy) and (**b**) ED (mSv) at helical and biopsy mode of CT-guided Iliac bone biopsy procedure.

**Figure 6 jimaging-09-00267-f006:**
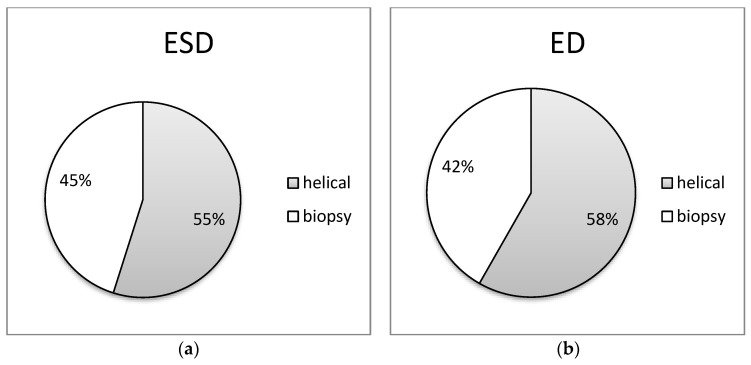
Distribution of (**a**) ESD (mGy) and (**b**) ED (mSv) at helical and biopsy mode of CT-guided liver biopsy procedure.

**Figure 7 jimaging-09-00267-f007:**
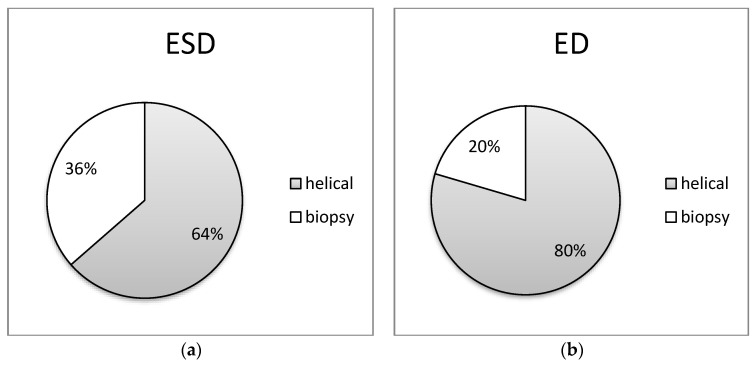
Distributions of (**a**) ESD (mGy) and (**b**) ED (mSv) at helical and biopsy mode of CT-guided lung biopsy procedure.

**Figure 8 jimaging-09-00267-f008:**
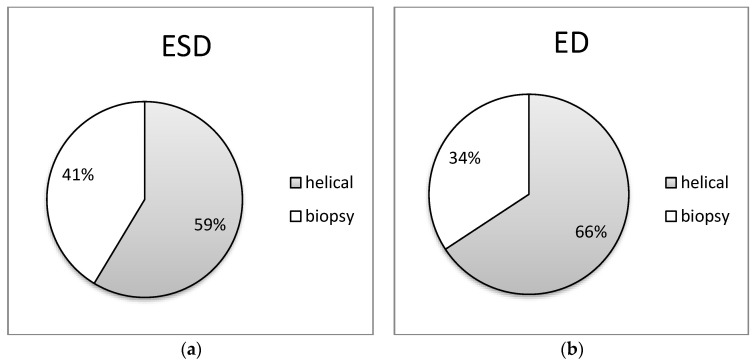
Distribution of (**a**) ESD (mGy) and (**b**) ED (mSv) at helical and biopsy mode of CT-guided mediastinum biopsy procedure.

**Figure 9 jimaging-09-00267-f009:**
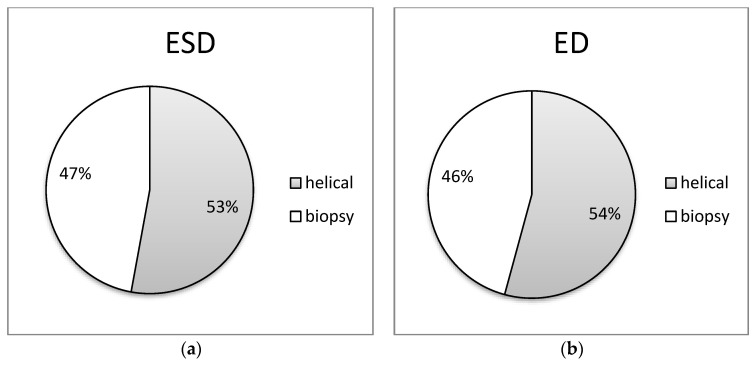
Distribution of (**a**) ESD (mGy) and (**b**) ED (mSv) at helical and biopsy mode of CT-guided para-aortic lymph nodes biopsy procedure.

**Figure 10 jimaging-09-00267-f010:**
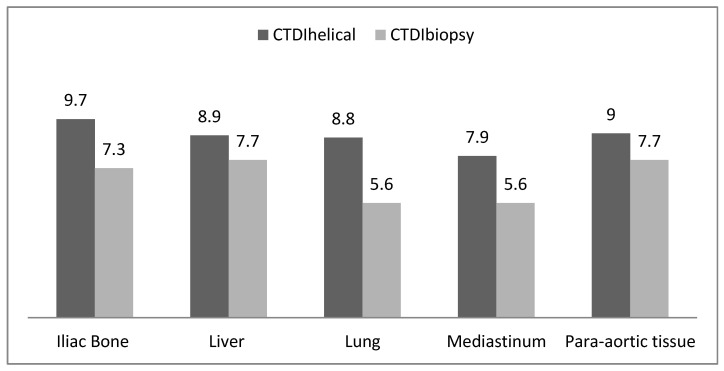
Typical DRL values in terms of CTDI_vol_ (mGy) at helical and biopsy scans for CT-guided biopsies.

**Table 1 jimaging-09-00267-t001:** Range of patient data and CT acquisition parameters of CT interventional procedures.

	Iliac	Liver	Lung	Mediastinum	Para-Aortic
Age (years)	(42–65)	(58–74)	(35–78)	(48–69)	(34–70)
Weight (kg)	(72–80)	(74–78)	(70–80)	(71–78)	(72–79)
kV	120	120	120	120	120
mAs	(50–150)	(60–180)	(65–180)	(55–250)	(70–220)
Collimation (mm)	(3–5)	(3–5)	(3–5)	(3–5)	(3–5)
N1	2	3	3	2	2
N2	5	5	4	5	6

N1: Number of helical acquisitions. N2: Number of helical biopsy acquisitions.

**Table 2 jimaging-09-00267-t002:** Overview of radiation dose descriptors for CT-guided iliac bone biopsy.

	Mean	SD	Min	25th Percentile	Median	75th Percentile	Max
CTDI_vol helical_ (mGy)	10.8	5.3	3.6	7.2	9.7	14.3	26.9
CTDI_vol biopsy_ (mGy)	7.5	2.6	3.2	5.5	7.4	8.6	14.9
DLP_helical_ (mGy cm)	400.4	427.8	67.0	185.5	263.3	516.8	2743.9
DLP_biopsy_ (mGy cm)	272.0	181.6	75.6	129.6	224.0	343.2	905.0
SSDE_helical_ (mGy)	14.8	6.8	6.2	10.0	12.6	20.1	36.3
SSDE_biopsy_ (mGy)	9.9	2.9	5.2	8.0	9.9	11.1	17.4
ED_helical_ (mSv)	6.0	6.4	1.0	2.8	3.9	7.7	41.2
ED_biopsy_ (mSv)	4.0	2.7	1.1	1.8	3.2	5.0	13.6
ESD_helical_ (mGy)	13.0	6.4	4.3	8.6	11.6	17.2	32.3
ESD_biopsy_ (mGy)	8.9	3.2	3.9	6.4	8.8	10.2	17.9

**Table 3 jimaging-09-00267-t003:** Overview of radiation dose descriptors for CT-guided liver biopsy.

	Mean	SD	Min	25th Percentile	Median	75th Percentile	Max
CTDI_vol helical_ (mGy)	10.1	4.1	4.4	7.3	8.9	12.0	23.8
CTDI_vol biopsy_ (mGy)	8.3	3.5	3.8	6.4	7.6	9.2	20.0
DLP_helical_ (mGy cm)	563.6	437.0	108.3	281.0	402.8	737.9	2417.0
DLP_biopsy_ (mGy cm)	403.9	800.5	44.8	149.9	246.0	387.0	6169.3
SSDE_helical_ (mGy)	13.2	4.7	6.4	9.8	11.8	16.5	27.6
SSDE_biopsy_ (mGy)	10.8	4.0	6.1	8.8	10.0	11.8	29.7
ED_helical_ (mSv)	8.5	6.6	1.6	4.2	6.0	11.1	36.2
ED_biopsy_ (mSv)	6.1	12.0	0.7	2.2	3.7	5.8	92.5
ESD_helical_ (mGy)	12.1	5.0	7.7	5.3	10.7	14.4	28.5
ESD_biopsy_ (mGy)	10.0	4.2	4.5	7.7	9.2	11.1	28.6

**Table 4 jimaging-09-00267-t004:** Overview of radiation dose descriptors for CT-guided lung biopsy.

	Mean	SD	Min	25th Percentile	Median	75th Percentile	Max
CTDI_vol helical_ (mGy)	10.5	5.9	3.2	7.8	8.8	12.4	47
CTDI_vol biopsy_ (mGy)	6.0	1.8	2.6	4.7	5.6	7.5	12.4
DLP_helical_ (mGy cm)	711.2	374.6	141.5	452.9	659.4	849.8	1938.8
DLP_biopsy9_ (mGy cm)	182.8	112.8	28.6	107.9	148.4	212.2	530.4
SSDE_helical_ (mGy)	14.7	11.9	4.9	10.4	12.0	16.5	97.8
SSDE_biopsy_ (mGy)	8.1	2.6	4.0	6.4	7.6	9.7	16.8
ED_helical_ (mSv)	10.0	5.2	2.0	6.3	9.3	11.9	27.1
ED_biopsy_ (mSv)	2.6	1.6	0.4	1.5	2.1	3.0	7.4
ESD_helical_ (mGy)	12.6	7.0	3.9	9.4	10.6	14.9	56.4
ESD_biopsy_ (mGy)	7.2	2.2	3.2	5.6	6.7	9.0	14.9

**Table 5 jimaging-09-00267-t005:** Overview of radiation dose descriptors for CT-guided mediastinum biopsy.

	Mean	SD	Min	25th Percentile	Median	75th Percentile	Max
CTDI_vol helical_ (mGy)	9.1	4.4	4.0	5.7	7.9	10.3	20.5
CTDI_vol biopsy_ (mGy)	6.4	2.6	3.7	4.6	5.6	7.1	15.1
DLP_helical_ (mGy cm)	489.6	578.6	102.1	154.4	388.6	613.7	3235.8
DLP_biopsy_ (mGy cm)	255.0	245.1	49.6	111.4	210.4	286.8	1236.5
SSDE_helical_ (mGy)	12.5	5.5	5.0	7.8	11.4	15.1	27.6
SSDE_biopsy_ (mGy)	8.7	2.8	5.0	6.6	7.8	10.0	16.3
ED_helical_ (mSv)	6.8	8.1	1.4	2.2	5.4	8.6	45.3
ED_biopsy_ (mSv)	3.6	3.4	0.7	1.6	2.9	4.0	17.3
ESD_helical_ (mGy)	10.9	5.3	4.8	6.8	9.5	12.3	24.6
ESD_biopsy_ (mGy)	7.7	3.1	4.5	5.5	6.7	8.5	18.1

**Table 6 jimaging-09-00267-t006:** Overview of radiation dose descriptors for CT-guided para-aortic tissue biopsy.

	Mean	SD	Min	25th Percentile	Median	75th Percentile	Max
CTDI_vohelical_ (mGy)	10.2	3.9	3.7	7.5	9.0	13.1	18.0
CTDI_volbiopsy_ (mGy)	9.1	5.0	3.5	6.3	7.7	10.4	25.0
DLP_helical_ (mGy cm)	641.3	592.4	107.5	254.8	453.4	797.9	2926.9
DLP_biopsy_ (mGy cm)	541.3	508.1	52.8	257.5	380.7	616.0	2546.7
SSDE_helical_ (mGy)	12.7	4.4	6.2	9.6	11.3	15.6	23.9
SSDE_biopsy_ (mGy)	11.0	5.0	5.9	8.2	9.4	12.5	24.2
ED_helical_ (mSv)	9.6	8.9	1.6	3.8	6.8	12.0	44.0
ED_biopsy_ (mSv)	8.1	7.6	0.8	3.9	5.7	9.2	38.2
ESD_helical_ (mGy)	12.2	4.7	4.5	9.0	10.7	15.7	21.6
ESD_biopsy_ (mGy)	10.9	6.0	4.2	7.6	9.2	12.4	29.9

## Data Availability

The data presented in this study are available on request from the corresponding author. The data are not publicly available due to ethical restrictions.

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
