# Peer review of "Patient Dose Estimation in Computed Tomography-Guided Biopsy Procedures"

_2313-433X, 2023, doi:10.3390/jimaging9120267_

Round 1
Reviewer 1 Report
Comments and Suggestions for Authors
Dear Authors,
I read with great interest your manuscript.
However, there are some aspects that require your attention.
Pay attention to referencing the text, the first reference appears after 1 page. This is unacceptable, although you feel that there is information already known you need to provide a reference at least at the end of each paragraph in the introduction.
Figures 5 to 9 present only percentages that could be summarized in text easily.
In the discussion section you need to address the high variability of thoracic mases. Already you have a bias by dividing the thorax into lung lesions and mediastinum. Underline the localization of the lesions deep or superficial at the level of the thorax. Obviously lesions easily accessible will reduce the duration of the procedure and the radiation dose. Reference this to the article by NeacÅŸu F, Vârban AÅž, Simion G, Åžurghie R, PătraÅŸcu OM, Sajin M, Dumitru M, Vrînceanu D. Lung cancer mimickers - a case series of seven patients and review of the literature. Rom J Morphol Embryol. 2021 Jul-Sep;62(3):697-704. doi: 10.47162/RJME.62.3.06. PMID: 35263397; PMCID: PMC9019611.
At the end of the manuscript insert the statements about ethics, author contributions and acknowledgements.
Please format the references according to MDPI instructions.
Looking forward to reading the improved version of the manuscript.
Author Response
- Comment from Reviewer #1:
Pay attention to referencing the text, the first reference appears after 1 page. This is unacceptable, although you feel that there is information already known you need to provide a reference at least at the end of each paragraph in the introduction.
Author response:
Thanks to the reviewer for this apposite remark. We have addressed this issue by adding the appropriate references (lines 34,37,45,51).
- Comment from Reviewer #1:
Figures 5 to 9 present only percentages that could be summarized in text easily.
Author response:
We understand the reviewer’s concerns. However, we believe that presenting the percentages in this way facilitates the reader to have a more comprehensive and illustrative view of the patient's radiation burden in the two different acquisitions modes.
- Comment from Reviewer #1:
In the discussion section you need to address the high variability of thoracic mases. Already you have a bias by dividing the thorax into lung lesions and mediastinum. Underline the localization of the lesions deep or superficial at the level of the thorax. Obviously lesions easily accessible will reduce the duration of the procedure and the radiation dose. Reference this to the article by NeacÅŸu F, Vârban AÅž, Simion G, Åžurghie R, PătraÅŸcu OM, Sajin M, Dumitru M, Vrînceanu D. Lung cancer mimickers - a case series of seven patients and review of the literature. Rom J Morphol Embryol. 2021 Jul-Sep;62(3):697-704. doi: 10.47162/RJME.62.3.06. PMID: 35263397; PMCID: PMC9019611.
Author response:
Thanks to the reviewer for this observation. We have underlined the significance of lesion’s localization to patients’ dose at the lines (566-568) We also added the proposed reference (reference No 40)
- Comment from Reviewer #1:
At the end of the manuscript insert the statements about ethics, author contributions and acknowledgements. Please format the references according to MDPI instructions.
Author response:
Thanks for pointing out these omissions. We have addressed this issue at the end of the manuscript. We also have modified the references according to MDPI instructions.
Reviewer 2 Report
Comments and Suggestions for Authors
This insightful article establishes Typical Diagnostic Reference Levels (DRLs) for patient doses in CT-guided biopsy procedures, analyzing 226 cases. The authors skillfully calculate key parameters like Effective Dose (ED) and Entrance Skin Dose (ESD), aligning with ICRP guidelines. Their work is significant in optimizing CT-guided biopsies, enhancing patient safety in radiological practices. The study's meticulous methodology and comprehensive data presentation make a notable contribution to clinical practices, promoting standardized radiation dose management in CT interventions. The authors' efforts in bridging clinical needs with radiation safety are commendable.
however, there are still some room to improve as below:
1,Broader Sample Size: While the study covers a diverse patient cohort, expanding the sample size and including multiple institutions could further validate the results.
2,Longitudinal Follow-Up: Including long-term follow-up data on patients could provide insights into the long-term effects of radiation exposure from these procedures.
3,Comparative Analysis: Comparing CT-guided biopsy techniques with other diagnostic modalities could offer a more comprehensive understanding of their relative efficacies and safety profiles.
Author Response
- Comment from Reviewer #1:
Broader Sample Size: While the study covers a diverse patient cohort, expanding the sample size and including multiple institutions could further validate the results.
Author response:
The sample size, as well as the fact that the patients concern one institution have been cited as limitations of this study and as a starting point for future studies. However, ICRP 135 instructions allow us the choice of one institution (typical values of DRL) and a number of patients (at least 20) per procedure.
- Comment from Reviewer #2:
Longitudinal Follow-Up: Including long-term follow-up data on patients could provide insights into the long-term effects of radiation exposure from these procedures.
Author response:
Thank you for this apposite remark. The institution follows a medical follow-up procedure for patients undergoing biopsy.
- Comment from Reviewer #2:
Comparative Analysis: Comparing CT-guided biopsy techniques with other diagnostic modalities could offer a more comprehensive understanding of their relative efficacies and safety profiles.
Author response:
This comment is very important; however the aim of the paper was to estimate the dose in patients undergoing CT biopsies, so the comparative assessment of patient doses was done based on similar studies. The evaluation of safety CT biopsy procedures in comparison with others biopsy, although it’s an important project, we consider to be beyond the aim of this paper.
Round 2
Reviewer 1 Report
Comments and Suggestions for Authors
Dear Authors,
You managed to answer all the objections from the reviewers.
Looking forward to receiving your future work.